# Current Landscape of Therapeutic Resistance in Lung Cancer and Promising Strategies to Overcome Resistance

**DOI:** 10.3390/cancers14194562

**Published:** 2022-09-20

**Authors:** Adnin Ashrafi, Zakia Akter, Pouya Modareszadeh, Parsa Modareszadeh, Eranda Berisha, Parinaz Sadat Alemi, Maria del Carmen Chacon Castro, Alexander R. Deese, Li Zhang

**Affiliations:** Department of Biological Sciences, The University of Texas at Dallas, Richardson, TX 75080, USA

**Keywords:** lung cancer, therapeutic resistance, chemotherapy, radiotherapy, targeted therapy, immunotherapy, tumor microenvironment, hypoxia

## Abstract

**Simple Summary:**

Despite an initial response to therapy, many lung cancer patients inevitably develop resistance to therapy leading to decreased duration of response and success of treatment. Recent research aims to elucidate mechanisms of resistance in order to improve drug response and treatment outcomes. By utilizing multidisciplinary approaches that target various resistance mechanism, it may be possible to delay development of treatment resistance or even resensitize cancers. This review aims to discuss novel approaches to improve clinical outcomes, delay the occurrence of resistance, and overcome resistance.

**Abstract:**

Lung cancer is one of the leading causes of cancer-related deaths worldwide with a 5-year survival rate of less than 18%. Current treatment modalities include surgery, chemotherapy, radiation therapy, targeted therapy, and immunotherapy. Despite advances in therapeutic options, resistance to therapy remains a major obstacle to the effectiveness of long-term treatment, eventually leading to therapeutic insensitivity, poor progression-free survival, and disease relapse. Resistance mechanisms stem from genetic mutations and/or epigenetic changes, unregulated drug efflux, tumor hypoxia, alterations in the tumor microenvironment, and several other cellular and molecular alterations. A better understanding of these mechanisms is crucial for targeting factors involved in therapeutic resistance, establishing novel antitumor targets, and developing therapeutic strategies to resensitize cancer cells towards treatment. In this review, we summarize diverse mechanisms driving resistance to chemotherapy, radiotherapy, targeted therapy, and immunotherapy, and promising strategies to help overcome this therapeutic resistance.

## 1. Introduction

Lung cancer is one of the leading causes of cancer deaths globally, with an approximately 1.8 million deaths per year [1]. It is a molecularly heterogeneous disease and can be divided into two major subtypes: non-small cell lung cancer (NSCLC) and small cell lung cancer (SCLC). NSCLC is further subdivided into histological subtype lung adenocarcinoma (LUAD), lung squamous cell carcinoma (LUSC), and large cell lung carcinoma (LCLC), which together account for 85% of all lung cancer cases. The remaining 15% of lung cancer is constituted by SCLC [2,3,4,5]. Widespread inter and intratumoral heterogeneity and transdifferentiation afford multiple mechanisms by which treatment resistance may develop and thus pose threats to the treatment of lung cancer [6]. Accurate subtyping is remarkably important for the treatment of lung cancer as major treatment options are determined based on histological examination performed by molecular pathology or immunohistochemistry.

Although cigarette smoking is a major risk factor, environmental pollution and genetic factors also contribute to the development of lung cancer. Cancer driver genes found in LUAD include Kirsten rat sarcoma viral oncogene homolog (KRAS), epidermal growth factor receptor (EGFR), tumor protein p53 (TP53 or p53), kelch-like ECH-associated protein 1 (KEAP1), serine threonine kinase 11 (STK11) also named as LKB1, and neurofibromatosis type 1 (NF1) [7]. Commonly mutated genes in LUSC are TP53, a tumor suppressor gene that is present in more than 90% of tumors, and cyclin dependent kinase inhibitor 2A (CDKN2A). Retinoblastoma (RB1) and p53 are genes usually mutated in SCLC [7,8]. The amplification of MYC family genes (MYC, MYCL, and MYCN), which are proto-oncogenes, is also observed in a subset of SCLC tumors [9]. Efforts towards classifying lung cancers based on genetic lesions have greatly helped to guide targeted therapies and improve clinical practice [10].

The treatment of lung cancer includes surgical resection, chemotherapy, immunotherapy, radiotherapy, and targeted therapy (Figure 1) [11,12]. There has been significant advancement in the treatment of lung cancer over the past two decades. The recent introduction of immune checkpoint inhibitors (ICIs), also named immune checkpoint blockades (ICBs), to target immune checkpoint proteins (ICPs) in the treatment of lung cancer patients, offers new promises [13]. Despite these advanced therapies, including immunotherapies, radiotherapies, and a combination of chemo-immunotherapies, resistance poses a challenge to clinical outcomes. Therapeutic resistance results in cancer recurrence and metastasis, and decreases patients’ lifespans [14]. Therefore, understanding the molecular mechanisms of therapeutic resistance is important in developing novel strategies to improve the overall survival (OS) rate of lung cancer patients. In this review, we discuss the mechanisms underlying the acquisition of resistance to different lines of therapy and potential strategies that can help overcome resistance.

## 2. Mechanism of Chemoresistance and Potential Therapeutic Inhibitors

Despite the growing interest in non-cytotoxic therapeutic agents, chemotherapy remains the standard treatment option for patients with advanced, unresectable NSCLC and first- and second-line management of SCLC [15,16,17]. There are four broad categories of chemotherapeutic drugs: (1) alkylating agents (e.g., platinum compounds such as cisplatin and carboplatin); (2) microtubule-targeting drugs (e.g., paclitaxel, docetaxel, and vinorelbine); (3) antimetabolites (e.g., pemetrexed and gemcitabine); and (4) topoisomerase inhibitors (e.g., etoposide) [18,19,20,21]. Despite the initial responsiveness of tumors to chemotherapies, lung cancer patients rapidly develop chemoresistance leading to disease progression.

Various factors contributing to chemoresistance (Table 1) include changes in drug influx and efflux, drug target alteration or inactivation, compartmentalization, epigenetic changes, and DNA damage. The blockade of cell-cycle arrest and apoptosis, the interaction of tumor microenvironments (TMEs), the acquisition of epithelial–mesenchymal evolution and cancer stem cell (CSC)-like phenotypes, unregulated microRNAs (miRNAs) expression, and metastasis are also responsible for chemoresistance [22,23,24,25,26]. TME is comprised of tumor and stromal cells such as fibroblasts, endothelial cells, and immune cells surrounded by a non-cellular component and vascular network. Cross-talk between the various components of the TME promotes tumor progression and therapeutic resistance by triggering hypoxia, a deficiency in nutrient supply, and vascular viscosity [27,28,29].

Studies show that upregulation of the DNA repair pathway mediates resistance to platinum-based chemotherapeutic agents. Platinum-induced DNA damage is repaired by two main pathways: nucleotide excision repair (NER) and homologous recombination (HR). Excision Repair Cross Complementing-1 (ERCC1) is a pivotal component of NER, whose expression is associated with cisplatin resistance in NSCLC [16,30,31].

The transportation of drugs is a controlled process that is mediated by the ATP-binding cassette (ABC) transporter family of proteins. Expression of the ABC superfamily of transport proteins, such as MRP1/ABCC1 and MRP3/ABCC3, was found to be correlated with cisplatin resistance [32]. Upregulated expression of ABCB1/MDR1/p-glycoprotein, ABCC3/MRP3, and ABCC10/MRP7 was observed in paclitaxel-, docetaxel-, or vinorelbine-resistant NSCLC cells [33,34,35,36,37,38].

Glutathione S-transferase (GST) isozymes, glutathione S-transferase pi 1 (GSTP1), and glutathione S-transferase alpha 1 (GSTA1) are predominantly overexpressed in cancer cells and play a significant role in the detoxification and inactivation of platinum drugs, followed by resistance to chemotherapy [22,39,40]. Upregulation of GSTP1 is linked with a poor response to anticancer drugs such as cisplatin. Glutathione S-transferase alpha 1 (GSTA1-1) overexpression appears to undermine the doxorubicin-dependent reduction in glutathione, particularly in the H69 SCLC cell line, by reducing lipid peroxidation [22,39,41]. Another research shows that inhibition of GSTP expression, through antisense cDNA, increases the cancer cell sensitivity to doxorubicin, cisplatin, and etoposide by reducing the detoxification of the drugs [22,39].

Associated with resistance to chemotherapy is the activation of signal transduction pathways such as EGFR and its downstream facilitators, which include the PI3K/Akt and mitogen-activated protein kinase (MAPK) signaling pathways, nuclear factor kappa-light-chain-enhancer of activated B cells (NF-κB), and signal transducers and activators of transcription 3 (STAT3) [42,43,44]. Ataxia-telangiectasia-mutated (ATM) is a member of the phosphatidylinositol 3-kinase- (PI3K) family of Ser/Thr protein kinases. NSCLC cells with acquired cisplatin resistance displayed an upregulation of ATM, phosphorylation/activation of its downstream effectors CHK2 and p53, and the overexpression of antiapoptotic proteins Bcl-2 and Bcl-XL. [45,46,47]. Pharmacological or genomic targeting of ATM, either alone or in combination with Mcl-1 (myeloid cell leukemia 1), which targets the B cell lymphoma 2 (Bcl-2) by acting as an antiapoptotic member of the Bcl-2 family of apoptosis-regulating proteins, restored sensitivity to cisplatin treatment in cisplatin-resistant cells [45,48]. Notably, Bcl-2 overexpression makes lung cancer cells resistant to apoptosis caused by DNA damage. Recent studies have shown that targeting Bcl-2-like proteins increases the effectiveness of platinum-based drugs, by eliminating cancer stem cells (CSCs) [49] as well as apoptosis-resistant cells [50,51,52,53]. As competitive inhibitors, antiapoptotic bcl-2 proteins [54] are currently under various clinical and preclinical stages of development for lung cancer treatment [53,55].

Downregulated expression of schlafen family member 11 (SLFN11), a member of the S-phase checkpoint, was found in the patient-derived xenograft (PDX) resistance model of SCLC [56,57]. SLFN11 is a relevant predictive biomarker of sensitivity to poly-ADP ribose polymerase (PARP) inhibitor monotherapy in SCLC, and its targeting can be a promising strategy to overcome resistance [57]. Inhibition of the cell cycle checkpoint kinase 1 (CHK1) was determined to have efficacy in platinum-resistant SCLC cells in vitro and in vivo [58,59]. Notch signaling is another important neuroendocrine stem cell signaling pathway. Cells with amplified Notch signaling are more chemoresistant than those undergoing more rapid expansion, suggesting notch signaling could enhance chemoresistance [60,61]. In a preclinical model, the inhibition of notch signaling in combination with chemotherapy increased the apoptosis of SCLC cells [60].

Geisslinger et al. demonstrated that lysosomal function interference might be an auspicious tactic in enabling sensitization to chemotherapy. Lysosomal function inhibition could target P-glycoprotein-driven chemoresistance, which might give rise to lysosome-targeted adjuvants in the future [62]. Zhan and colleagues successfully developed a new autophagy inhibitor, alpha-hederin, that changes lysosomal pH and inhibits the lysosomal maturation of cathepsin D [22]. The buildup of undigested material leads to the accumulation of reactive oxygen species (ROS), consequently enhancing its killing effect on tumor cells [22,63]. These mechanisms make alpha-hederin an effective agent in overcoming resistance to paclitaxel in NSCLC [64]. Additionally, the induction of sphingosine kinase-1 (SphK1) by insulin-like growth factor 1 (IGF1) was found to increase the tolerance of NSCLC cells to paclitaxel treatment, while pharmacological suppression of SphK1 restored paclitaxel sensitivity [65]. Similarly, epigenetic modifications play an important role in the regulation of DNA-mediated events, including transcription, DNA repair, and replication [66]. Aberrant regulation of such events promotes tumor formation, progression, and chemotherapy resistance. For example, the hypermethylation of the *IGFBP3* promoter, and upregulation of forkhead box F1 (FOXF1) expression was found to trigger resistance to cisplatin and the acquisition of cancer stem cell (CSC)-like phenotypes in NSCLC cells [67].

Furthermore, increased expression and activation of WNT/b-catenin signaling was found to cause chemoresistance [68,69,70,71,72]. One in vitro study demonstrated that a protein kinase c (PKC) inhibitor, GF109203X, inhibits WNT5A-induced cell migration, invasion, and clonogenicity in A549 and A549/DDP (diammine dichloro platinum) lung cancer cells. This signifies the role of WNT5A in promoting lung cancer cell movement through WNT/PKC non-canonical pathway activation [73,74]. Gardner et. al. reported that the gene coding for twist family transcriptional factor (TWIST1) was upregulated in chemoresistant cells [75]. This enhanced TWIST1 expression is correlated with acquired drug resistance, epithelial-to-mesenchymal transition (EMT), metastasis, and stemness [76,77,78,79]. TWIST1 is a WNT-inducible transcription factor that relates to the dysregulation of WNT signaling detected in clinical samples and SCLC cell lines [80]. Therefore, targeting molecules involved in the WNT pathway could be a promising strategy to overcome chemotherapy resistance. Cisplatin or cis-diamminedichloroplatinum(II) (CDDP) resistance can be stimulated by variations in numerous intracellular pathways involving miRNAs. MiRNAs are a family of non-coding RNAs that regulate gene expression via sequence-specific targeting of mRNAs [81,82]. During the process of lung cancer development, miRNAs can operate as tumor suppressor genes or oncogenes. Hua and his colleagues found that overexpression of one such miRNA, miR-1, improved CDDP sensitivity of NSCLC cells through ATG3-mediated autophagy inhibition, providing a possible therapeutic target for reducing chemoresistance [83]. miR-106a, miR-31, miR-15b, miR-27a, miR-223, miR-205, miR-92b, and miR-224 promote resistance to cisplatin by downregulating the expression of adenosine triphosphate-binding cassette transporter A1 (ABCA1), adenosine triphosphate-binding cassette subfamily B member 9 (ABCB9), phosphatidylethanolamine-binding protein 4 (PEBP4), Raf kinase inhibitory protein (RKIP), F-box/WD repeat-containing protein 7 (FBXW7), phosphatase and tensin homolog (PTEN), and p21, respectively [84,85,86,87]. Altogether, studies show that targeting dysregulated miRNAs may be a promising strategy for enhancing therapeutic effectiveness, or overcoming drug resistance in lung cancer patients [88,89]. Other mechanisms of chemoresistance include metabolic reprogramming, communication between tumor cells and the surrounding TME, and the alteration of microtubules [38,90,91,92,93]. A better understanding of interactions mediating chemotherapy resistance will allow the discovery of novel targets to delay resistance onset and/or resensitize tumors to existing treatments.

## 3. Mechanism of Radiotherapy Resistance in Lung Cancer

Radiotherapy (RT) is one of the main lung cancer treatments and is effective for almost half of all cancer patients [106,107]. RT includes internal radiation therapy (brachytherapy) and external beam radiation therapy (EBRT). EBRT includes 3-D conformal radiation therapy (3-D CRT), stereotactic body radiation therapy (SBRT), proton therapy, prophylactic cranial irradiation (PCI), hypofractionated radiation therapy, and intensity-modulated radiation therapy (IMRT) [108,109,110,111,112,113,114,115,116]. Despite the use of RT for lung cancer treatment, unfortunately, the therapeutic outcomes are not always satisfactory, and tumor radioresistance can lead to the reduction in the efficiency of RT, resulting in tumor recurrence and metastasis [106,117]. Thus, it is necessary to investigate the molecular and cellular mechanisms responsible for the loss of radiosensitivity and discover potential therapeutic targets that might help overcome radioresistance.

### 3.1. Targeting Signaling Pathways Associated with Radioresistance and Potential Strategies for Radiosensitization in Lung Cancer

Diverse pro-survival and metastatic signaling pathways mediate cancer cells’ survival [117]. Genetic alterations of the *PI3K*, *AKT*, *PTEN*, *EGFR*, and *KRAS* genes, MET amplification, and EML4-ALK rearrangements are associated with lung cancer progression [118]. Moreover, radiation is linked to the dysregulation of these pathways such as the hyperactivation of PI3K/AKT, which induces RAC expression and activity, increases levels of EMT markers, and promotes invasive phenotypes and metastasis [117,119]. Furthermore, AKT may also contribute to radioresistance by promoting double-strand DNA break (DSB) repair [119]. Additionally, upregulation of gelsolin, a protein involved in cytoskeleton remodeling, is associated with the activation of PI3K/AKT signaling and promotion of radioresistance in NSCLC cells. Cancer cells overexpressing gelsolin showed reduced apoptosis, and decreased levels of cleaved caspase-3 and PARP after irradiation [120]. Diverse studies have focused on targeting these signaling pathways to overcome radioresistance. For example, downregulation of AKT and ERK via vascular endothelial growth factor receptor 2 (VEGFR2) inhibition can enhance radiosensitivity by increasing radiation-induced G2/M phase arrest and inhibiting radiation-induced DSB repair in NSCLC cells [121]. mTOR-dependent expression of hypoxia inducible factor 1 α (HIF1-α) was also linked to proliferation and cancer cell survival under hypoxic conditions. Inhibition of mTOR can decrease HIF1-α expression and restore radiation sensitivity in lung adenocarcinoma cells [122]. In addition, inhibiting RAC1, PI3K, MEK, and AKT overexpression can enhance radiation sensitivity by blocking DSB repair [117,119].

Cancer cells maintain a high redox level reflected by an increased ROS production and an activated antioxidant defense system equilibrium, which promotes neoplastic growth. Low and moderate levels of ROS can induce cancer cells’ proliferation and survival by post-transcriptional modifications of diverse proteins and kinases [123]. RT anticancer effects depend on the promotion of ROS accumulation, which causes cytotoxic oxidative stress [124]. However, radiation exposure may increase ROS and mitochondrial dysfunction, which are associated with pro-survival pathways and radio-adaptive resistance. Thus, ROS can mediate the cytotoxic effect of RT but also regulate the pro-survival adaptive response and induce radioresistance mainly due to the redox compensatory mechanisms in cancer cells. While ROS-mediated RT may eliminate the majority of cancer cells, the enhanced antioxidant capacity of cancer cells may allow some of them to survive the elevated ROS environment and induce cancer relapse and progression [124,125]. Low-dose ionizing-radiation can promote elevated ROS production inducing autophagy and activation of the Nrf2-HO-1 antioxidant pathway in lung adenocarcinoma cells [126]. Moreover, mutations in Keap1 are associated with Nrf2 constitutive activation in cancer cells and radioresistance. Nrf2 is also involved by redirecting glucose and glutamine to the serine, glutathione, and purine nucleotides synthesis pathways [127]. Thus, targeting glutaminase inhibition can radiosensitize KEAP1 mutant cells [128]. Furthermore, several studies have focused on plant-derived compounds due to their antineoplastic potential and low toxicity. For instance, ferulic acid was reported for its antioxidant and anti-inflammatory activities, and for its protective effect against gamma radiation-induced DNA damage. Ferulic acid treatment prior to gamma radiation increased already elevated ROS levels, inhibited the ROS-PI3K/Akt-p38 MAPK-NF-kB-MMP-9 pathway, and supported mitochondrial apoptotic pathway activation and cell cycle progression inhibition in cancer cells promoting their radiosensitization [123]. Therefore, targeting ROS can constitute a potential strategy to overcome radioresistance in lung cancer.

Lung cancer cells can acquire radioresistance through a mechanism of EMT [129]. Transcription factors Snail, Slug, and twist-related protein 1 (TWIST1) are frequently upregulated during the transition, with associated changes to EMT-related biomarkers such as E-cadherin, N-cadherin, and vimentin [78,130,131,132]. Cadherins such as E-cadherin and N-cadherin are components of adherens junctions between cells, along with catenins such as β-catenin. During EMT, the expression of N-cadherin in place of E-cadherin, along with the increased expression of intermediate filament protein vimentin, results in decreased cell–cell adhesion and changes in adhesion to the extracellular matrix, altering regulatory pathways such as contact inhibition and aiding in migration [133]. Understanding the underlying regulatory mechanisms of the EMT offers promising new targets to prevent or reverse its effects and overcome radioresistance. For example, β-Elemene, an inhibitor of the Prx-1/NF-kB/iNOS signaling pathway, radio-sensitized radioresistant NSCLC cells, decreased expression of N-cadherin and vimentin, and increased E-cadherin expression [134,135]. In another study, inhibition of chemokine (C-X-C motif) receptor 4 (CXCR4) in NSCLC decreased radioresistance, while overexpression increased it [136]. CXCR4 interacted with STAT3, an activator of Slug [136,137]. Inhibition of tescalcin (TESC), a protein in the TESC/c-Src/IGF1R signaling pathway, interfered with EMT and radio-sensitized NSCLC [138]. TESC may increase the expression of aldehyde dehydrogenase isoform 1 (ALDH1) via the activation of STAT3 [138]. RAD001, an inhibitor of the mTORC1 signaling pathway, inhibited EMT and radio-sensitized NSCLC [139]. E-cadherin expression levels increased while the expression of vimentin decreased when mTORC1 signaling was inhibited [139]. Treatment with EG00229, an inhibitor of neuropilin 1 (NRP1) binding to VEGF, radio-sensitized adenocarcinoma cell lines and reduced N-cadherin and vimentin, while NRP1 overexpression increased them. NRP1′s effect on EMT could result from the PI3K/AKT/mTOR, IL6/STAT3, or SDF-1/CXRC4 signaling pathways [140]. The silencing of C2 domain-containing phosphoprotein (CDP138) overcame radioresistance in NSCLC, and targeting growth differentiation factor 15 (GDF15) signaling, which may alter TGF-β/Smad signaling, resulting in increased Snail expression and EMT [141]. TWIST1 expression made NSCLC more resistant to radiation, while treatment with F-Box and Leucine Rich Repeat Protein 14 (FBXL14) destabilized TWIST1, radio-sensitizing the treated cells. FBXL14 expression levels had a negative correlation with EMT markers [142]. Taken together these studies suggest that targeting regulatory pathways associated with Snail, Slug, and TWIST1 to interfere with EMT could prevent radioresistance.

### 3.2. miRNAs in Radioresistant Lung Cancer Cells

miRNAs participate in many cellular function and is associated with radiotherapy process [143,144]. However, the deregulation of miRNAs expression can promote radioresistance in NSCLC by inhibiting the essential functional proteins involved in the radiation treatment response [145,146]. In recent studies focused on the role of miRNAs in lung cancer radioresistance, miR-145 was found to enhance radiosensitivity in NSCLC by targeting TMOD3 mRNA. Overexpression of miR-145 [147] was suggested as a potential strategy to increase the effectiveness of RT and sensitize radioresistance in lung cancer cells. Similarly, overexpressing miRNA-9 and miR-328-3p can enhance the sensitivity to RT in NSCLC cells [148,149]. Likewise, upregulation of miRNA-320a acts as an inhibitor of radioresistance in NSCLC by suppressing HIF1α and increasing PTEN methylation [144].

A study conducted in 2021 by Xue et al. clarified that miR-129-5p transfection into NSCLCs induces apoptosis, cell cycle arrest, and DNA injury, and can radio-sensitize NSCLCs by targeting SOX4 and RUNX1 [150]. Overexpressed FOXO3 is the target for miR-182 to increase radioresistance. Therefore, knocking down miR-182 can result in cell cycle arrest without changing the DNA damage repair system [151]. In addition, knocking down oncomiRNA, miR-410, can reduce DNA damage repair by targeting PTEN/PI3K/mTOR [152]. Overall, targeting different miRNAs in lung cancer appears to be a promising approach to overcome radioresistance.

### 3.3. DNA Damage Associated with Radioresistance in Lung Cancer

DNA damage is one of the major causes of radioresistance in lung cancer [153]. Ubiquitin-specific protease 14 (USP14) acts as a regulator [154] in double strand breaks (DSBs), which affects both non-homologous end-joining (NHEJ) and HR [155,156]. Knocking down USP14 by using short hairpin RNA (shRNA) showed a notable increase in the radiosensitivity of NSCLCs [154].

In another study, upregulation of serine proteinase inhibitor clade E member 2 (SERPINE2) and Sirtuin 3 acts as a regulator of radiosensitivity in lung cancer and is directly involved in the DNA repair mechanism by facilitating the phosphorylation of HR-mediated DSBs’ repair. Knocking down SERPINE2 in radioresistant cells made them more radiosensitive [157,158]. Moreover, knocking down Sirt3 can arrest the cell cycle and activate the ATM-Chk2 pathway over irradiation [158]. Transfecting an integrin beta-1 (ITGB1) short hairpin RNA (shRNA) increased radiation-induced DNA damage and arresting of the G2/M phase. The downstream effector of ITGB, Yes-associated protein 1 (YAP1), is suppressed by ITGB and it can induce radioresistance by affecting DNA repair, while inhibiting ITGB1 can help with radiosensitivity [159]. Moreover, activation of the A2B receptor promotes the recovery of irradiated lung cancer cells from DNA damage by the mediation of the γ-radiation-induced translocation of EGFR and phosphorylation of src and EGFR, while A2B inhibition is a therapeutic approach to make cancer cells radiosensitive [160]. Therefore, learning more about DNA damage-related mechanisms and pharmacologically inhibiting them can be a potential strategy to overcome radioresistance in lung cancer cell lines.

## 4. Targeted Therapy Resistance in Lung cancer and Overcoming Strategies

Targeting oncogenic driver alterations remains a potent and effective therapeutic approach, especially for patients with NSCLC subtypes [161,162]. EGFR and ALK inhibitors have become the mainstays of treatment in lung cancer therapy, while in recent years a new class of drugs has been able to target the once “un-druggable” KRAS mutation [163,164,165]. While targeted therapies might initially be effective for patients, chronic drug exposure leading to further alterations in these oncogenic drivers significantly reduces treatment efficacy [166]. In this section, we will discuss common resistance mechanisms to targeted therapies and strategies to overcome treatment resistance.

### 4.1. Overcoming EGFR TKI Resistance

More than 30% of all NSCLC patients have EGFR positive tumors, and patients with EGFR positive tumors are overwhelmingly female, non-smokers, and Asian [167]. Tyrosine kinase inhibitors (TKIs) that target EGFR mutations are, at least initially, an effective treatment for these patients. These include the first-generation reversible EGFR TKIs gefitinib and erlotinib, the second-generation irreversible TKIs afatinib and dacomitinib, and the third-generation irreversible TKI osimertinib [168].

Resistance to EGFR TKIs can be on-target (also known as EGFR-dependent) or off-target (also known as EGFR-independent) [169]. The most common acquired resistance mechanism following first-line treatment with first- or second-generation EGFR TKIs is the T790M mutation [170]. However, the T790M mutation does not affect responsiveness to osimertinib, a third-generation TKI [170]. Interestingly, osimertinib is effective both as a second-line treatment for previously treated EGFR positive tumors with acquired T790M mutation and as a first-line treatment for EGFR positive, advanced NSCLC [171]. The most common resistance mechanism to osimertinib is acquired C797S mutation [172], which spells the end of effective treatment with available EGFR TKIs for patients.

Notably, allosteric kinase inhibitors represent a fourth generation of EGFR TKIs that can overcome osimertinib resistance. JBJ-04-125-02, an allosteric kinase inhibitor, is a L858R-specific mutant-selective allosteric EGFR inhibitor that shows efficacy against EGFR positive C797S mutant NSCLCs as a single-agent in both in vitro and in vivo models [173]. The use of JBJ-04-125-02 in combination with osimertinib enhanced the binding ability of JBJ-04-125-02 and increased antitumor efficacy in mouse models [173]. This combination limited EGFR-dependent resistance mechanisms and delayed therapeutic resistance in vitro and in vivo [173]. JBJ-04-125-02 is among a new class of allosteric inhibitors that hold promise for overcoming C797S mutants [173].

Novel mechanisms by which EGFR resistance may be delayed and/or EGFR mutants may be resensitized to TKIs have also been explored. Targeting GRB2, an EGFR-binding adaptor protein, alongside the use of the EGFR TKI, icotinib, has been shown to delay resistance to EGFR TKIs in cancer cell lines and mouse models. Lymecycline, a derivative of the antibiotic tetracycline, was shown to target GRB2 and reverse resistance to icotinib in NSCLC cell lines. In mouse models, the addition of lymecycline to icotinib treatment produced a synergistic effect with no significant increase in toxicity [174]. Additionally, the enzymatic activity and expression of EHMT2 are upregulated in NSCLC cells [175]. The inhibition of EHMT2, a histone lysine methyltransferase, was shown to restore erlotinib sensitivity in resistant cancer cells. Combination of an EHMT2 inhibitor and erlotinib further enhanced antitumor effects in an EGFR TKI-resistant NSCLC mouse model [175].

EGFR-independent resistance includes MET amplification and HER2 amplification [169]. Normally expressed by stem and progenitor cells, MET amplification correlates with poor prognosis. HER2, also a common biomarker for breast cancer, is part of the EGFR family and can mediate resistance to TKIs as well. The HIF-1 inhibitor YC-1 was shown to resensitize the human LUAD cell line (HCC827) with acquired resistance with MET amplification to gefitinib, a first-generation EGFR TKI. The HIF-1 pathway allows cancer cells to survive in hypoxic conditions through transcriptional activation of the genes needed for survival and growth [176].

Additionally, insights into the emergence of EGFR TKI resistance present possible therapeutic targets. Heme levels were found to be elevated in osimertinib-resistant EGFR-mutant NSCLC cell lines and in the blood plasma of osimertinib-treated EGFR-mutant NSCLC patients [177]. Notably, plasma heme levels were most elevated in patients experiencing progression-free survival (PFS) of less than 15 months, as opposed to those having PFS > 15 months [177]. Our lab has previously shown that elevated heme synthesis and import underpin tumorigenic functions, and that heme sequestration effectively suppresses tumor growth and progression in NSCLC mouse models [178,179,180]. Thus, it is possible that heme plays a role in EGFR-dependent resistance pathways and targeting heme can be a viable strategy to overcome this resistance.

### 4.2. Overcoming ALK Drug Resistance

Anaplastic lymphoma kinase (ALK) gene rearrangements are a major hallmark of lung cancer. Aberrations in this gene, which codes for a protein tyrosine kinase, promote cancer proliferation and survival. The fusion of ALK with echinoderm microtubule-associated protein like-4 (EML4) gene creates the fusion gene EML4-ALK, which is the most observed ALK fusion in lung cancer [181,182]. ALK mutations are mutually exclusive with EGFR and KRAS mutations in many patients, positioning ALK as the primary target for treatment in ALK-positive lung cancers [183].

ALK inhibitors posed a breakthrough advancement in NSCLC treatment, as the standard first-line treatment was systemic chemotherapy before advancements in ALK inhibitors [184]. The first generation ALK inhibitor crizotinib, receiving accelerated approval by the FDA in 2011 for metastatic NSCLC, showed a median of 7.7 months PFS versus 3.0 months in chemotherapy controls pemetrexed or docetaxel in a phase III clinic trial A8081007 [185,186]. While this posed a major improvement in the treatment of ALK-positive tumors, most patients went on to develop resistance 1–2 years after treatment [187]. Primary resistance is characterized as a refractory response to initial treatment, likely attributed to different fusion variants [187]. Secondary resistance, or acquired resistance, falls into ALK dominant and ALK non-dominant categories. In ALK dominant secondary resistance, there is a mutation that prevents inhibition of the targeted tyrosine kinase. In ALK non-dominant resistance, there is a development of other mutations, such as EGFR, KRAS, or KIT amplification, which bypasses ALK [187].

The second generation ALK inhibitor ceritinib was developed as an oral ALK inhibitor 20 times more potent than crizotinib [188]. Ceritinib is effective for patients with developed resistance against crizotinib and was thus first approved for refractory NSCLC after crizotinib treatment [187]. Another second-generation ALK inhibitor, alectinib, displayed activity against L1196M mutations, which leads to resistance against crizotinib [189]. It was also approved for patients with progression on or after crizotinib. Similarly, brigatinib was developed as an additional second-generation ALK inhibitor with similar efficacy against L1196M mutations, as well as EGFR T790M mutations [187].

However, resistance soon emerged against these next generation TKIs, so third-generation lorlatinib was developed as an ALK and ROS1-inhibitor designed to target the driver mutations of crizotinib and second-generation ALK inhibitors [184]. In clinical trials, lorlatinib displayed greater PFS compared to crizotinib, positioning itself as a preferred first-line treatment against ALK-positive NSCLC [190,191]. Ensartinib, a second-generation small molecule ALK inhibitor, was created to target central nervous system (CNS) metastases [192,193]. Entrectinib, approved for neurotrophic tyrosine receptor kinase (NTRK) and ROS-1 NSCLC in 2019, targets ALK aberrations and can additionally cross the blood–brain barrier [192].

While second- and third-generation ALK inhibitors helped to overcome single mutation ALK resistance, double mutant ALK resistance seems inevitable [194]. Fourth-generation ALK TKIs are being developed that target double ALK-resistant mutations to help overcome drug resistance. Even so, double and triple mutation may develop against 4G ALK TKIs, but would still likely achieve a minimum of 35–40 months of PFS [194]. Off-target resistance mechanisms such as EGFR TKIs, including parallel bypass mechanisms such as MET amplification or RET rearrangement, or downstream signaling pathways such as BRAF fusions and MAP2K1 mutations, among several others, may occur [195]. Other off-target resistance mechanisms such as histological transformations to SCLC or EMT have also been observed after treatment with both EGFR and ALK TKIs, although the exact molecular mechanisms are unclear [195].

### 4.3. Strategies to Overcome Resistance to KRAS G12C Inhibitors

KRAS is the most common genetic alternation in NSCLC, occurring in 20–40% of LUADs [196]. KRAS is part of the RAS family of GTPases that play important roles in cell proliferation and survival, with KRAS being the most common RAS mutation in cancer constituting 85% of RAS mutations in cancer [197]. While KRAS has been an attractive target for anticancer therapies, it has remained elusive as attempts to inhibit KRAS have been hindered by the high affinity of RAS to bind GTP as well as a smooth structure with no obvious pockets for inhibitor binding. After many failures, two KRAS inhibitors have shown exciting results against the G12C KRAS variant, which occurs in nearly half of NSCLC KRAS mutations: sotorasib (Amgen) and adagrasib (Mirati Therapeutics). KRAS G12C inhibitors bind in the switch II pocket, locking KRAS into the inactivated state [198]. In clinical trial “KRYSTAL-1” in NSCLC, adagrasib showed a median 12.6-month OS, indicating strong clinical efficacy, with similar results observed for sotorasib in the “CodeBreaK100” study showing a median OS of 12.5 months [199,200].

The emergence of resistance is inevitable, as seen by preclinical and clinical data, due to high inter and intratumoral heterogeneity in lung cancer [201]. There are three main molecular mechanisms of resistance: on-target mechanisms, off-target mechanisms, and histological transformation [202]. Treatment with adagrasib was shown to increase acquired KRAS mutations or amplify bypass mechanisms such as MET amplification. In NSCLC, RAS-MAPK activation is a bypass mechanism due to mutations in the RAS-RAF-MEK-ERK pathway [202]. Another possible strategy for acquired resistance to KRAS G12C inhibitors could be to switch between sotorasib or adagrasib, although this may be ineffective for Y96D and Y96S mutations [202]. Cell line studies in BA/F3 cells showed that using BI-3406, a SOS1 inhibitor, or TNO 155, a SPH2 inhibitor, could overcome resistance against these mutations when used with KRAS G12C inhibitors [202].

Mitochondrial targeting could additionally serve as a strategy to overcome KRAS resistance. Apoptosis-inducing factor (AIF) deletion in KRAS G12D mouse lung cancer has been shown to lead to increased survival by leading to decreased oxidative phosphorylation (OXPHOS) and increased glycolytic activity [203]. This evidence is corroborated in human NSCLC patients where AIF expression has been shown to be associated with worse outcomes, as well as other studies showing the importance of mitochondrial activity and OXPHOS in both NSCLC and SCLC tumor progression [180]. In KRAS-driven colorectal cancers, mitochondrial inhibitors, such as tigecycline, have been shown to reduce tumor growth in vivo, which suggests that this combination may yield positive results in lung cancers [204]. This evidence suggests that co-targeting of mitochondria may be an effective strategy in overcoming KRAS resistance.

## 5. Mechanism of Immunotherapeutic Resistance and Overcoming Strategies

Cancer immunotherapy is one of the most attractive therapeutic options over traditional therapies including chemotherapy, targeted therapy, and RT because of its promising clinical responses [205]. It targets the host immune system to make it fight against cancer cells and possesses promise to restore antitumor immunity [206]. Currently, various types of immunotherapies including adoptive T-cell therapy, cancer vaccine, ICIs, and cytokine modulators are approved by the Food and Drug Administration (FDA) against different types of cancer [207,208]. The use of ICIs directed against ICPs has showna stable response, long-term survival benefits, and increased PFS [209,210,211] in NSCLC patients. It is now considered as the first line of treatment alone or in combination with surgery, chemotherapy, and RT in patients with NSCLCs [212]. The FDA has approved immunotherapeutic drugs named as nivolumab, pembrolizumab, atezolizumab, and durvalumab to treat NSCLC patients [213]. These drugs, known as ICIs, inhibit PD-1 expression on the surface of T cells and PD-L1 overexpression on tumor cells, preventing the binding between PD-1/PD-L1, and restoring antitumor immunity [213]. However, resistance to ICIs either as primary or acquired resistance is commonly found in many patients [214,215] based on PD-L1 expression, where patients do not show any response to ICIs or develop resistance after 8–10 months of clinical benefit, respectively. Unfortunately, reduced expression of PD-L1 and major histocompatibility complex (MHC) proteins, lower levels of effector T cells, and a higher number of immune suppressor cells represent limited antitumor responses to ICIs in SCLC patients [216]. Therefore, it is necessary to study and target these resistance mechanisms that are summarized in Table 2 to resensitize cancer cells to immunotherapy and develop novel approaches for overcoming immunotherapeutic resistance.

### Targeting Immunoresistance Mechanism to Resensitize Lung Cancer Cells to Immunotherapy

Gene mutations, modified TME, tumor mutational burden (TMB), the DNA mismatch repair (MMR) pathway, and altered epigenetics are reported as major drivers for modifying immunotherapeutics (Table 2) and promoting immunotherapy resistance [215,217,218]. Inadequate tumor immunogenicity associated with a low TMB, gene mutation, the lack of antigen presentation and PD-L1 expression, and mutations in the PI3K-Akt, IFN, and Wnt/B-catenin signaling pathways are considered as tumor intrinsic immunoresistance mechanisms, while immunosuppressive TME, epigenetic modifications, and the expression of alternative immunocheckpoints are associated with tumor extrinsic immunoresistance mechanisms [215].

It has been found that mutations in cancer driver genes, *KRAS*, *EGFR*, *ALK*, human epidermal growth factor receptor 2 (*HER2*), *STK11*, *CDKN2A*, *TP53*, etc. [7], dysregulate the expression of the PD-1 and PD-L1 and PD-1/PD-L1 signaling pathways, which are associated with resistance in immunotherapy [219,220,221]. The reversing of [219] STK11 mutations by knocking down the *STAT3* gene was shown to improve the anti-PD-L1 and anti-cytotoxic T-lymphocyte-associated antigen (CTLA-4) resistance in NSCLCs. Different types of mutated signaling pathways including Wnt/B-catenin, JAK/STAT3, PI3K-Akt, interferon gamma (IFN-γ) signaling pathways, and mutations in Janus kinase (JAK1/2) accelerate resistance to anti-PD-1 [222], RT plus anti-CTLA4 [223], anti-CTLA4 [224], and anti-PD-1 [225] therapy, respectively, which are essentially needed to be considered for overcoming resistance to ICIs [215]. The upregulated PI3K-Akt signaling pathway reduces the function of CD8+ T cells by promoting the recruitment of immunosuppressive cells into TME, the expression of VEGF, and PD-L1 expression [215].

Alterations in the MMR system lead to the formation of microsatellite instability (MSI), short tandem repeat sequences, which are highly immunogenic and display high TMB neoantigen [241]. TMB, which is defined as the total number of somatic mutations found in the genome of cancer cells, works as an independent predictor for analyzing the outcome of the treatment to ICIs [242]. According to FDA, TMB ≥ 10 is considered—a good predictor of the immunotherapy response. A higher mutation burden in the genome of cancer cells leads to the increased possibility of producing neoantigens that can be targeted by ICIs to neutralize tumor cells [242]. CD8+ tumor-infiltrating lymphocytes (TIL) are reported to be elevated in MMR-deficient (dMMR) cancer and this dMMR shows a positive impact such as a stable response, OS, and PFS on immunotherapy [241,243]. This positive correlation was also observed between dMMR and the response to ICIs from a retrospective analysis conducted in NSCLC patients treated with nivolumab, which suggests the analysis of the MMR system improves immunotherapy [244].

Various cells found in TME such as tumor-associated macrophages (TAMs), regulatory T cells (Tregs), B regulatory cells (Bregs), cancer-associated fibroblasts (CAFs), and myeloid-derived suppressor cells (MDSCs) suppress immune responses and exhibit resistance to immunotherapy in many ways [228]. For instance, these cells block the activity of effector CD4+ and CD8+ T cells and release immunosuppressive molecules and angiogenic factors, such as transforming growth factor beta (TGF-β), interleukin 10 (IL-10), prostaglandin E2, and VEGF, respectively. These cells upregulate the expression of ICPs such as PD-1, PD-L1, and CTLA-4 as well as the expression of alternative ICPs, lymphocyte-activation gene 3 (LAG-3), and T-cell immunoglobulin mucin-3 (TIM-3). These events consequently trigger immunotherapeutic resistance [228,233,234] by turning immunosupportive TME into immunosuppressive TME.

Neo-angiogenesis, the formation of new blood vessels by tumor cells, is one of the hallmarks of cancer that enhances cancer cell growth, proliferation, and metastasis. The emergence of tumor vasculature from existing blood vessels affects regular blood flow, oxygen levels, and nutrient levels in TME [245]. This neo-vasculature induces the expression of angiogenic factors and VEGF, creates hypoxia, and recruits immunosuppressive cells (Treg, MDSC, CAF, and TAM) into TME, which causes immunosuppressive TME [246]. It has been found that angiogenic factors upregulate downstream signaling pathways involved in cancer cell proliferation, reduce adhesion molecules leading to the inhibition of the infiltration of effector T cells into TME, and impair the delivery of therapeutic agents into tumor cells [229]. Thus, the inhibition of neo-angiogenesis is promising to amplify the efficacy of immunotherapeutic agents.

Notably, cancer cells depend on enhanced mitochondrial energetic metabolism for ensuring high energy and oxygen to support their growth, proliferation, and metastasis [247]. High oxidative metabolism accompanied with oxygen consumption [247] creates hypoxia, which, in turn, activates HIF-1 signaling pathways, affects vasculature by triggering the release of VEGF, increases immunosuppressive cells, and reduces effector T cells [231]. All these events triggered by hypoxia modify TME, reduce immunotherapeutic efficacy, and trigger resistance to ICIs [231,232]. Huang et al. targeted mitochondria using atovaquone, which inhibited the expression of OXPHOS genes and lowered granulocytic-MDSCs and Treg cells [230]. This approach increased the number of tumor infiltrating CD4+ T cells and accelerated responses to anti-PD-1 therapy in lung cancer. Interestingly, heme sequestering peptide 2 (HeSP2) was shown to reduce OXPHOS levels and angiogenesis and improve hypoxia by normalizing HIF1A, VEGFA, and VEGFR1 in in vivo mouse models [179,180]. Therefore, targeting high oxidative phosphorylation and hypoxia might be a potential strategy for overcoming immunotherapeutic resistance.

Deregulated epigenetics promote immunotherapeutic resistance through modifying the expression of immune related genes and triggering immune suppressive phenotypes and dysfunctional T cells [215,218]. In a study conducted in NSCLC patients with cancer progression, the combination of Entinostat (ENT), a histone deacetylase inhibitor, and pembrolizumab showed synergistic and antitumor effects in phase II trials compared to anti-PD-1 treatment alone [248]. Therefore, using epigenetic inhibitors in combination with ICIs could be promising strategies for overcoming limitations in immunotherapy in lung cancer.

Multiple types of circular RNAs (circRNAs), for instance, hsa_circ_0000190 [236], hsa_circ_0079587 [237], circFGFR1 [238], circUSP7 [239], etc., have been found to be associated with promoting tumor progression, metastasis, immune evasion, and immunotherapy resistance [235]. These circRNAs provoke resistance to anti-PD-1 therapy by upregulating PD-L1 expression, targeting CXCR4, and inhibiting the infiltration of CD8+ T cells, which supports the necessity of targeting circRNAs to resensitize tumor cells to immunotherapy.

Gut microbiota composition is one of the factors that affect responses to immunotherapy and make tumor cells resistant to ICIs [249,250]. It worked as a good predictor to analyze the immunotherapeutic efficacy towards anti-PD-1 treatment in a study conducted on Chinese NSCLC patients [251]. Lung cancer patients, having low levels of the bacterium *Akkermansia muciniphila* in the gut due to antibiotic consumption, showed resistance to anti PD-1 treatment [240,249]. Huang et al., used ginseng polysaccharides (GPs), which is an extract of Panax ginseng, in combination with αPD-1 monoclonal antibody (mAb) to target gut microbiota in lung cancer mice models. They found that this combination significantly enhanced effector T cells and reduced Foxp3+ regulatory T cells, resulting in an improved antitumor response [240]. Therefore, targeting gut microbiota and analyzing its composition could be a promising approach for enhancing the efficacy of anti-PD-1/anti-PDL1 and overcoming resistance to immunotherapy in lung cancer cells.

## 6. Combination Approaches for Improving Therapeutic Resistance and Future Prospects

Although there have been many improvements in the treatment options for lung cancer, the evolution of resistance in therapy challenges the clinical outcomes of patients and reduces OS [16]. Recent advances in immunotherapy have revolutionized the landscape of clinical therapy in NSCLC. The FDA has already approved ICIs, anti–PD-1 and anti–PD-L1 antibodies, for the treatment of NSCLC. Despite the advancement in clinical outcomes, patients eventually fail to respond to immunotherapy due to the emergence of primary or secondary resistance. Therefore, combinations of immunotherapy with chemotherapy drugs or treatment combinations with radio and targeted therapy have been explored for better treatment outcomes, which are mentioned in Table 3. Given that immunotherapy drugs, targeted therapy drugs, and chemotherapy drugs act on different targets and cells, synergistic or combined treatment of these therapies or irradiation may achieve greater therapeutic effects at the cost of lower or similar side effects (Figure 2). Currently, a rising number of clinical trials are in progress to further explore new regimens as monotherapy or in the combination of chemotherapy with molecular targeted therapies, including first-, second-, and third-generation EGFR-TKI and ICIs [15,104].

Chemotherapy resistance can be overcome by following combination approaches with other therapies. Studies manifest that the addition of pembrolizumab to chemotherapy (carboplatin and paclitaxel or nab-paclitaxel) extends life expectancy in metastatic squamous NSCLC patients [252]. In one of the clinical trials of NSCLC, the IMpower150 trial, the addition of atezolizumab to bevacizumab plus chemotherapy significantly improved PFS and OS among patients with metastatic NSCLC, regardless of PD-L1 expression and EGFR or ALK genetic alteration status [253].

Immunotherapy-resistant lung cancers can be resensitized through combinations of ICIs and chemotherapeutic drugs. Cytotoxic chemotherapy drugs induce cell death and cause changes in the TME, enhancing antigen presentation, Tregs’ and MDSCs’ activity abrogation, and increasing T-cell activation and infiltration [254], etc. In a phase 3 trial involving metastatic NSCLC, a combination of anti-PD-1, pembrolizumab, and chemotherapy drugs, pemetrexed and cisplatin/carboplatin, resulted in an OS of 69.2% compared to the placebo–chemotherapy combination’s OS of 49.4% at the 12-month mark [255].

Combinations between ICIs and anti-angiogenesis agents are another potential approach, serving to simultaneously target an immunosuppressive TME and tumor neo-angiogenesis, to potentiate immunotherapy [214]. As mentioned earlier, neo-angiogenesis or tumor vasculature is an extrinsic resistance mechanism in immunotherapy-resistant cancers that plays a significant role in promoting an immunosuppressive TME and preventing the delivery of immunotherapeutic agents [228]. In a phase II clinical trial involving a combination of anti-PD-1 (avelumab), and the receptor kinase inhibitor drug (axitinib) targeting VEGFR showed an objective response rate (ORR) of 31.7% in chemotherapy-resistant and recurrent NSCLC [260]. Therefore, targeting VEGF and ICPs, following combination approaches, could serve as an effective strategy to overcome immunotherapy resistance in lung cancer.

One of the promising strategies for overcoming primary and acquired immunotherapy resistance is through the synergistic effects of PD-1/PD-L1 and CTLA-4 inhibitors. These synergistic effects are due to anti-PD-1/anti-PD-L1 acting on the early phases of immune activation, while anti-CTLA-4 acts on the later phases, in the peripheral tissues [263]. For instance, in combination between anti-PD-1 (nivolumab) and anti-CTLA-4 (ipilimumab), the OS rate increased, compared to nivolumab monotherapy alone [261]. These combinations help in the regulation of T-cell priming and activation [254] to overcome resistance. Furthermore, alternative ICPs such as LAG-3, TIM-3, and T-cell immunoglobulin and ITIM domain (TGIT) are upregulated due to ICI inhibition, leading to T-cell exhaustion and eventually adaptive resistance [228]. Combinations between common ICI’s and alternative checkpoint inhibitors have been shown to cause resensitization.

Combination therapy is also a promising strategy to overcome resistance to EGFR TKIs. This approach aims at evading drug resistance through a so-called bypass signaling mechanism by targeting other parallel pathways. Preclinical studies have demonstrated that upfront treatment of EML4-ALK-positive lung tumors with both an ALK inhibitor and a MAPK pathway inhibitor can substantially postpone or even inhibit the onset of resistance [264]. Similar findings were detected in EGFR-mutant lung adenocarcinoma preclinical models and individuals with BRAF V600E lung adenocarcinoma [265,266]. In mouse models, a combination of carboplatin chemotherapy and sotorasib in KRAS G12C lung cancer resulted in greater tumor regression than either monotherapy [267]. Adagrasib combination therapy with palbociclib, a CDK4/6 inhibitor, yielded similar results [268]. A combination of anti-PD-1 immunotherapy with both sotorasib and adagrasib showed complete tumor regression in mice with KRAS G12C tumors [267]. This may be due to the inhibition of KRAS leading to less immunosuppressive TME as KRAS signaling affects IL-10 and TGF-β [269].

Another resensitization method is through ICI and RT combinations. RT functions by causing DNA damage, which eventually leads to cell death. Irradiation creates an immune response through cyclic GMP–AMP synthase (cGAS) stimulation of interferon gene (STING) (cGAS-STING) pathway activation, upregulating the type 1 interferon [270], which causes resensitization through MHC I upregulation [271]. It leads to antigen visibility through T-cell priming, alteration to the TME, etc., [270]; however, it also upregulates PD-L1 expression [272]. A combination results in synergistic effects to potentially overcome immunotherapy resistance by enhancing the effects of immunotherapy through RT [270]. In PEMBRO-RT, treatment with anti-PD-1 (pembrolizumab) after stereotactic body RT showed an increase in ORR [262]. This trial indicates that a combination can be effective in overcoming resistance.

Cisplatin-resistant NSCLC cells display upregulation of peroxisome proliferator-activated receptor gamma (PPARγ) and coactivator-1 alpha (PGC-1α). PGC-1α is a transcription factor coactivator that promotes mitochondrial respiration, heme biosynthesis, angiogenesis [105], and OXPHOS with downregulation of glycolysis [273,274], causing hypoxia by inducing HIF-1α [103,275]. Similarly, hypoxic stress and changes in TME are known to cause radioresistance and effect immune effector cells such as CD8+ and CD4+ T cells [276,277]. Many studies have confirmed that some cancer cells depend on OXPHOS and targeting OXPHOS and mitochondrial respiration overcome their resistance [278,279,280,281]. The author’s lab extensively studies the relationship of heme with lung cancer and has generated HeSP2 that significantly alleviates tumor hypoxia and normalizes tumor vasculature, raising the possibility of their combination with chemotherapy drugs such as cisplatin or immune therapy assisting in improving antitumor efficacy [282].

To optimize the outcome of combination therapy, another strategy of nanotechnology-based co-encapsulation and co-delivery can be used. This will ensure equal spatial and sequential drug dissemination in the target tumor cells [283]. This method can help overcome resistance [284] as it will enhance drug stability by increasing concentrations of drug accumulation in tumor cells with lower side effects.

## 7. Conclusions

Drug resistance is a major issue in lung cancer that leads to treatment failure, tumor progression, and relapse. Recently, the significant improvement in understanding the complex landscape of therapeutic resistance has changed the treatment paradigm of lung cancer. For instance, platinum-based drug treatment in combination with first-, second-, and third-generation EGFR TKI (erlotinib, afatinib, and osimertinib, respectively), ALK TKIs (ceritinib), or ICIs (pembrolizumab, nivolumab, and atezolizumab) has improved the clinical benefits and patients’ lifespan. Although the initial response is remarkable, the patient ultimately develops resistance. Hence, targeting multiple mechanisms such as deregulated TME, hypoxia, upregulation of alternative ICPs, immunosuppressive TME, deregulated autophagy, DNA damage, and EMT could work as a promising strategy to overcome therapeutic resistance and to improve clinical outcomes in lung cancer patients.

Altogether, combination therapy that targets multidisciplinary approaches, is a great strategy to delay the onset of resistance. However, optimal doses and timings of the administration of such combinations alongside toxicity levels are yet to be determined. Although there are several ongoing clinical trials, many exhibit optimistic results, while others are inconclusive, such as the results found in the PEMBRO-RT phase 2 randomized clinical trial. [262]. In this regard, translational research is important alongside defining the immune patterns of patients, or screening for TME characteristics, tumor cell immune phenotypes, specific mutations, and the host’s immune status. It will also help in developing the best strategies in terms of dose, timing, duration, and sequence of administration, thus reducing toxicity. Therefore, the identification of novel therapeutic methods that target the potential markers, mediators, or pathways triggering therapeutic resistance is necessary to develop precision medicine to reduce lung cancer progression, increase OS, and prevent the occurrence of resistance.

## Figures and Tables

**Figure 1 cancers-14-04562-f001:**
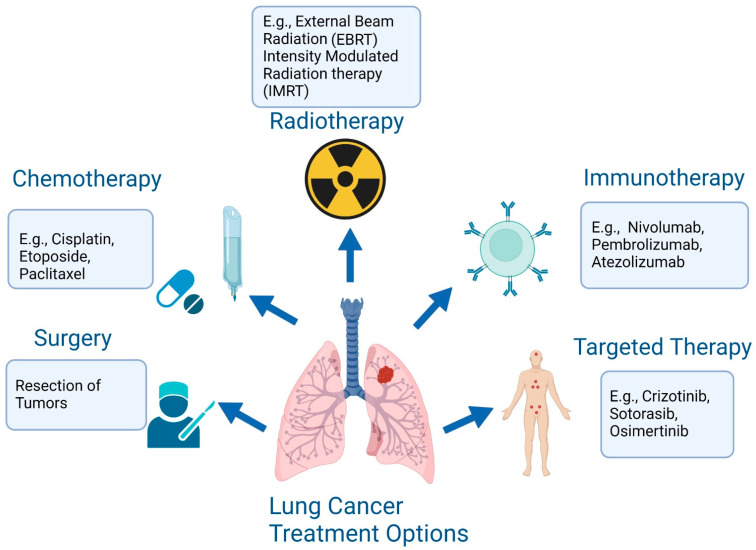
The treatment options for lung cancer are surgery, chemotherapy, radiotherapy, immunotherapy, and targeted therapy (created with BioRender.com (accessed on 1 September 2022).

**Figure 2 cancers-14-04562-f002:**
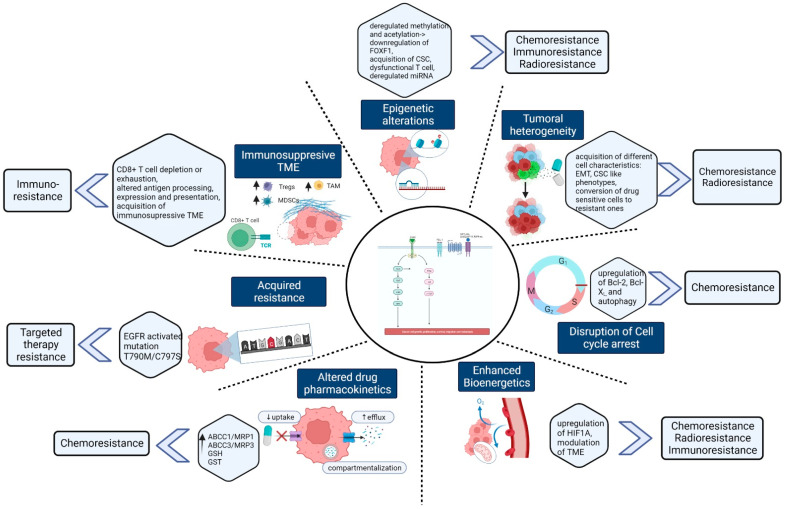
Different mechanisms inducing therapeutic resistance in lung cancer: Therapeutic resistance against chemotherapy, radiotherapy, targeted therapy, and immunotherapy in lung cancer is caused by different types of mechanism. For instance, tumor heterogeneity, alteration in drug influx and efflux, compartmentalization, epigenetic changes, hypoxia, or reduced autophagy stimulate chemoresistance in lung cancer. Radioresistance is found to happen by epithelial and mesenchymal transition, DNA damage, dysregulated miRNA, and changes in various signaling pathways. Mutations in EGFR or KRAS genes and targets as well as alterations in drug sensitivity lead to mutations at T790M that cause primary and acquired resistance for targeted therapy. Mutations in cancer driver genes, immunosuppressive TME, modified epigenetics, and high bioenergetic are mainly responsible for triggering immunoresistance in lung cancer (created with BioRender.com (accessed on 15 September 2022).

**Table 1 cancers-14-04562-t001:** Summary of mechanism of chemotherapeutic resistance in lung cancer.

Mode of Action	Target Entity	Chemotherapeutic Agent	References
DNA repair system	Upregulation of: ERCC1 DNA Polymerase	Platinum compounds	[94,95]
Drug efflux	Upregulation of ABC family transporters: ABCC1/MRP1 ABCC3/MRP3 ABCB1/MDR1 ABCC10/MRP7 ABCB1/MDR1/p-glycoprotein ABCC6/MRP6 ABCC11/MRP8	Platinum compounds Microtubule-targeted compounds Etoposide Gemcitabine Pemetrexed	[33,34,35,36,37,38]
Prosurvival signaling	Upregulation of: EGFR PI3K/Akt MAPK Calpain Sphk1	Platinum compounds Microtubule-targeted compounds Pemetrexed	[42,43,44]
Cell cycle arrest	Upregulation of: Bcl-2 Bcl-XL Autophagy	Platinum compounds Gemcitabine Microtubule-targeted compounds	[46,47,96]
Epigenetic regulation	Promoter methylation of IGFBP3 and FOXF1 Upregulation of KDM3B Deregulation of circadian rhythm	Platinum compounds	[67,97,98,99]
MicroRNA	Upregulation of: miR-106a, miR- 31, miR-15b, miR-27a, miR-223, miR-205, miR-92b, miR-224, miR-34c-5p, miR-181a, miR-135a, miR 197-3p, miR-222-3p downregulation of: miR-101-3p, miR-181, miR-589, miR-1244, miR-29c, miR-630, and miR-197 miR-16, miR-17-5p, miR-216b, miR-200b, miR-363-3p	Platinum compounds Microtubule-targeted compounds Etoposide Gemcitabine	[16,84,85,86,87]
EMT/CSC	Upregulation of: EMT phenotype Notch signaling Wnt signaling Shh signaling	Platinum compounds Microtubule-targeted compounds Etoposide Pemetrexed	[100,101]
Tumor microenvironment	Upregulation of: Hypoxia CAF PDL-1	Platinum compounds	[102,103,104]
Cancer metabolism	Upregulation of: PGC1α and glutamine metabolism Downregulation of: OXPHOS and glycolysis	Platinum compounds	[90,105]

**Table 2 cancers-14-04562-t002:** Summary of mechanism of immunotherapeutic resistance and prospective targets in lung cancer.

Resistance Mode	Action	Target	References
Gene mutations	a. Modify the expression of PD-1, PD-L1, and CTLA-4 proteins b. Deregulate PD-1/PD-L1 signaling pathways	ALK, EGFR, HER-2, CDK2NA, STK11 (LKB1); TP53; KRAS	[219,221,226,227]
Dysregulation of cellular and molecular pathways	a. Promote primary and adaptive resistance to anti-CTLA-4 and anti-PD-1 b. Enhance cancer cell proliferation and metastasis	Wnt/B-catenin, JAK/STAT3, PI3K-Akt, JAK1/2 mutations, IFN-γ signaling pathways	[215,222,224]
Neo-angiogenesis	a. Inhibit the infiltration of effector immune cells b. Upregulate the expression of PD-L1 c. Recruit Treg, TAM, and MDSC cells d. Impair the delivery of therapeutic agents to tumor cells e. Reduce adhesion molecules into TME	Hypoxia, HIF1-A, VEGFA, VEGFR, Angiopoietin-2 (ANG2)	[228,229]
High oxidative metabolism	a. Trigger hypoxia b. Promote cancer cell growth, proliferation, and metastasis c. Create immunosuppressive TME d. Affect effector T cells	OXPHOS complexes, heme, HIF1-A, VEGFA, VEGFR	[179,180,230,231,232]
Immunosuppressive TME	a. Recruit immunosuppressive cells (Tregs, Bregs, MDSCs, TAM, and CAF) b. Inhibit the infiltration of the effector immune cells c. Release of proinflammatory molecules d. Upregulate immune checkpoint proteins (ICPs) e. Affect antitumor immunity	Alternative ICPs, immunosuppressive molecules, proinflammatory molecules, VEGFA, HIF1-A	[228,233,234]
Upregulation of alternative immune checkpoints	a. Modulate TME and show adaptive resistance to anti-PD-1	LAG-3, TIGIT, TIM3, and TIM-1	[228,234]
Deregulated epigenetics	a. Modify the expression of immune related genes b. Triggers T-cell dysfunction	DNA methyl transferase; histone methyl transferase; histone deacetylase	[215,218]
Dysregulated circRNAs	a. Upregulate PD-L1 b. Recruit inflammatory molecules c. Inhibit the CD8+ T cells’ infiltration into tumorigenic regions	hsa_circ_0000190, hsa_circ_0079587, circFGFR1, circUSP7	[235,236,237,238,239]
Modified gut microbiota	a. Reduce effector T cells b. Increase regulatory T cells c. Affect antitumor responses	Gut microbiota composition; gut bacteria	[240]

**Table 3 cancers-14-04562-t003:** An outline of clinical trials following combination approaches in lung cancer patients.

Drug Combination	Phases/Study	Treatment Outcome	References
Platinum + pemetrexed + pembrolizumab	Phase III (Keynote 189)	median OS—22.0 months; median PFS—9.0 months	[256]
Carboplatin + (nab)-paclitaxel + pembrolizumab	Phase III (Keynote 407)	median OS—15.9 months; median PFS—6.4 month	[257]
Carboplatin + nab-paclitaxel + atezolizumab	Phase III (Impower 130)	median OS—18.6 months; median PFS—7.0 months	[258]
Carboplatin + paclitaxel + bevacizumab + atezolizumab	Phase III (Impower 150)	median OS—19.2; median PFS—8.3 months	[253]
Pembrolizumab + platinum + pemetrexed	Phase III	median OS—12 months; median PFS—8.8 months	[255]
Nivolumab + ipilimumab + two cycles of chemotherapy	Phase III (CheckMate 9LA)	median OS—15.6 months;	[259]
Avelumab + axitinib	Phase II (Javelin Medley VEGF)	ORR—31.7%; median PFS—5.5 months	[260]
Nivolumab + ipilimumab	Phase III (CheckMate 227)	median OS—17.1 months	[261]
Pembrolizumab + stereotactic body radiation therapy (SBRT)	Phase II (PEMBRO-RT)	median OS—15.9 months; median PFS—6.6 months	[262]

Abbreviations: OS—overall survival; PFS—progression-free survival; ORR—objective response rate.

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
