# Peer review of "Current Landscape of Therapeutic Resistance in Lung Cancer and Promising Strategies to Overcome Resistance"

_cancers, 2022, doi:10.3390/cancers14194562_

Round 1
Reviewer 1 Report
The review aims to discuss the main mechanisms of therapy resistance in lung cancer together with new strategies to overcome them. The topic is certainly broad and worthy of interest, and there is currently no such comprehensive review on the subject.
However, if this is a strength, it is at the same time the main weakness of the manuscript, which can be published, but after careful review by the authors.
Indeed, the attempt to cover such a large number of topics within the limits required of a review makes it difficult to understand at times. The manuscript is superficial, with limited insights into biological mechanisms that are often only listed. Although the problem is present throughout the manuscript, it is section 2 that suffers most. In fact, within it, the mechanisms of chemoresistance are summarily listed and lack adequate depth.
My suggestion is to revise the various sections, devoting a few clarifying sentences to the individual mechanisms listed. In addition, discussing them on the basis of a common criterion (e.g. grouping mechanisms that act on the same signalling pathway or impairs the same therapy) could facilitate reading and understanding of the important information reported.
It is clear that such a revision of the manuscript has a major impact on its length. However, it might be preferable for the review to focus on fewer aspects, but in a comprehensive manner, rather than covering all areas at the expense of clarity.
Author Response
We greatly appreciate the reviewers’ meticulous review of our manuscript and constructive comments. To address the points raised in the comments, we have now extensively revised the manuscript. These additions and revisions include the following major improvements: (1) Mechanism of chemotherapy resistance (section 2) were elaborately explained and reorganized, (2) Table-1 was included in chemotherapy section to summarize the key factors and targets involved on the mechanism of chemoresistance (3)Additional clarifying sentences were added in all the sections and revised if necessary according to the reviewer’s suggestion.
Below is a description of our responses to each comment (the original comments are quoted verbatim):
Comment: The review aims to discuss the main mechanisms of therapy resistance in lung cancer together with new strategies to overcome them.  The topic is certainly broad and worthy of interest, and there is currently no such comprehensive review on the subject. 
However, if this is a strength, it is at the same time the main weakness of the manuscript, which can be published, but after careful review by the authors.
Indeed, the attempt to cover such a large number of topics within the limits required of a review makes it difficult to understand at times. The manuscript is superficial, with limited insights into biological mechanisms that are often only listed. Although the problem is present throughout the manuscript, it is section 2 that suffers most. In fact, within it, the mechanisms of chemoresistance are summarily listed and lack adequate depth.
 My suggestion is to revise the various sections, devoting a few clarifying sentences to the individual mechanisms listed. In addition, discussing them on the basis of a common criterion (e.g. grouping mechanisms that act on the same signaling pathway or impairs the same therapy) could facilitate reading and understanding of the important information reported.
It is clear that such a revision of the manuscript has a major impact on its length. However, it might be preferable for the review to focus on fewer aspects, but in a comprehensive manner, rather than covering all areas at the expense of clarity.
Revision: Thank you for your valuable comments. We have revised the chemoresistance section (section 2) and mentioned the major mechanisms in a comprehensive manner. We have included a table (Table-1) to integrate the key and mostly studied targets for the mechanism of chemoresistance. Likewise, we have included many clarifying sentences in all the sections wherever it was necessary.
Reviewer 2 Report
In this review, the authors summarized the molecular mechanisms that lead to therapy resistance in lung cancer. It’s a significant effort and comprehensively covered multiple topics including chemotherapy, radiotherapy, targeted therapy, and immunotherapy. However, the reviewer found some of the languages are vague and shared mechanisms(mutations) were discussed in a reciprocal way between sections that seem redundant. The reviewer has suggested some modifications that may make this review more concise and informative as listed below.
*Figure 2, authors need to be more specific about how each mechanism contributes to resistance in different types of therapy. Eg. drug efflux is less relevant to radiotherapy and immunotherapy. In addition, it’s unclear how acquired mutations in oncogenes and epigenetic changes contribute to resistance eg. making lung tumor cells more proliferative/less apoptotic, etc. What is presented here is much generic that applies to all cancer, authors are supposed to make a demonstration specific to lung cancer eg. indicate some lung cancer specific pathway/genes.
*line 102-117, does the author wish to discuss MYC regulated targets? If so, the authors would need to mention how mTOR and CHEK1 are related to MYC. Or, if the authors wish to talk about pathways rewired in resistance, it would be better to include one sentence at the beginning of this paragraph saying something like “pathway rewiring has been found chemo-resistant lung cancer cells”. Otherwise, it’s very confusing.
*Authors named the altered genes/pathway yet did not establish their function to resistance. For instance, what specific role do these genes in Line 102-132 play in molecular biology and how do they impact cancer cell fitness? A proper presentation would be, that BCL2 is an antiapoptotic protein> BCL2 overexpression makes lung cancer cells resistant to apoptosis caused by DNA damage.
*Line 148-167, literature cited here does not seem to support the “use of nanoparticles and ADC” at the beginning of this paragraph. Eg. antisense cDNA to target GSTs.
*table1 is a good summary. May consider adding tables in other sections. And include ongoing clinical trials.
*Conclusion should be discussing the most promising agents that may overcome resistance in each section.
Line 62, serious clinical consequences are vague
Line 64, prognosis and overall survival are redundant
Line 66-67, need to be concise
Line 78, use response instead of sensitivity; disease progression is a consequence of chemoresistance
Line 88, extra space?
Line 102, a set genes are directly bound by MYC and associated with.
Line 103, simulate chemosensitivity if awkward. Consider using might re-sensitize SCLC and NSCLC, or restore chemosensitivity.
Line 104, use “for example” instead of “likewise”. Also, need to discuss how mTOR pathway (upstream, cap-dependent translation) is related to MYC otherwise readers may find it difficult to follow.
Line 136-137, the way authors describe the role is miRNA is too vague
Line 141, word of choice “easing”
Line 153, what does the accumulation of ROS do and how is it related to killing tumor cells?
Line 332, tumor-inhibiting capability, additive effect, or synergistic?
Line 333, rephrase, inhibition of EHMT2, a histone lysine methyltransferase, was shown
Line 335, “activity” is too broad, be specific eg. kinase activity, binding kinetics. May move 335 to line 332 as an introduction
Lin 434, at the end of this section, may discuss the off-target resistance of these inhibitors. Check out Alissa J.Coopers et al. recent review on this topic in nature reviews clinical oncology
Line 399, no need to use KRAS full spelling here as it was first mentioned in line 44
Author Response
We greatly appreciate the reviewers’ meticulous review of our manuscript and constructive comments. To address the points raised in the comments, we have now extensively revised the manuscript. These additions and revisions include the following major improvements: (1) Mechanism of chemotherapy resistance (section 2) were elaborately explained and reorganized, (2) Table-1 was included in chemotherapy section to summarize the key factors and targets involved on the mechanism of chemoresistance (3)Additional clarifying sentences were added in all the sections and revised if necessary according to the reviewer’s suggestion.
Below is a description of our responses to each comment (the original comments are quoted verbatim):
Comment: In this review, the authors summarized the molecular mechanisms that lead to therapy resistance in lung cancer. It’s a significant effort and comprehensively covered multiple topics including chemotherapy, radiotherapy, targeted therapy, and immunotherapy. However, the reviewer found some of the languages are vague and shared mechanisms(mutations) were discussed in a reciprocal way between sections that seem redundant. The reviewer has suggested some modifications that may make this review more concise and informative as listed below.
*Figure 2, authors need to be more specific about how each mechanism contributes to resistance in different types of therapy. Eg. drug efflux is less relevant to radiotherapy and immunotherapy. In addition, it’s unclear how acquired mutations in oncogenes and epigenetic changes contribute to resistance eg. making lung tumor cells more proliferative/less apoptotic, etc. What is presented here is much generic that applies to all cancer, authors are supposed to make a demonstration specific to lung cancer eg. indicate some lung cancer specific pathway/genes.
Revision: Thank you for your suggestions. We have revised figure 2 according to your comments.
Comment: *line 102-117, does the author wish to discuss MYC regulated targets? If so, the authors would need to mention how mTOR and CHEK1 are related to MYC. Or, if the authors wish to talk about pathways rewired in resistance, it would be better to include one sentence at the beginning of this paragraph saying something like “pathway rewiring has been found chemo-resistant lung cancer cells”. Otherwise, it’s very confusing.
Revision: Thank you for your suggestions. We realized that it might be better if we include MYC regulated proteins in targeted therapy (section 4) part and discussed there accordingly. To avoid having redundancy we deleted that part from the mechanism of chemoresistance (section 2).
Comment: *Authors named the altered genes/pathway yet did not establish their function to resistance. For instance, what specific role do these genes in Line 102-132 play in molecular biology and how do they impact cancer cell fitness? A proper presentation would be, that BCL2 is an antiapoptotic protein> BCL2 overexpression makes lung cancer cells resistant to apoptosis caused by DNA damage.
Revision: Thank you for your suggestions. We have revised the entire section 2. We have discussed the genes and pathways and their specific targets whose upregulation or downregulation causes resistance. We have included the relevant information about Bcl-2.
Comment: *Line 148-167, literature cited here does not seem to support the “use of nanoparticles and ADC” at the beginning of this paragraph. Eg. antisense cDNA to target GSTs.
Revision: Thank you for your suggestion. We have deleted the part “use of nanoparticles and ADC” from section 2.
Comment: *table1 is a good summary. May consider adding tables in other sections. And include ongoing clinical trials.
Revision: Thank you for your suggestion. We have included a table (Table-1) into section 2 summarizing the major targets involved in the mechanism of chemoresistance. Ongoing clinical trials have already been listed into table-3 that summarizes promising combination approaches following chemotherapy, immunotherapy, and radiotherapy.
Comment: *Conclusion should be discussing the most promising agents that may overcome resistance in each section.
Revision: Thank you for your suggestion. We have rewritten the conclusion and mentioned promising targets to overcome resistance. We discussed specific combination drugs or agents in section 6 and Table 3.
Comment: Line 62, serious clinical consequences are vague
Revision: Thank you. We have deleted it.
Comment: Line 64, prognosis and overall survival are redundant
Revision: Thank you. We have deleted “prognosis” to avoid redundancy.
Comment: Line 66-67, need to be concise
Revision: Thank you. We have reduced the wording to make it more concise.
Comment: Line 78, use response instead of sensitivity; disease progression is a consequence of chemoresistance
Revision: Thank you. We have changed the word “sensitivity” to “response” as suggested.
Comment: Line 88, extra space?
Revision: Thank you. We have fixed the extra space.
Comment: Line 102, a set genes are directly bound by MYC and associated with.
Revision: Thank you for your suggestion. We have revised and reorganized section 2. Therefore, to avoid redundant information, we have deleted this particular sentence as myc-associated genes are discussed in section 4.
Comment: Line 103, simulate chemosensitivity if awkward. Consider using might re-sensitize SCLC and NSCLC, or restore chemosensitivity.
Revision: Thank you. We have changed the word to re-sensitize as suggested.
Comment: Line 104, use “for example” instead of “likewise”. Also, need to discuss how mTOR pathway (upstream, cap-dependent translation) is related to MYC otherwise readers may find it difficult to follow.
Revision: Thank you for your suggestion. We have revised and reorganized section 2. Therefore, to avoid redundant information, we have deleted this sentence and MYC related genes are discussed in section 4.
Comment: Line 136-137, the way authors describe the role is miRNA is too vague
Revision: Thank you for your comment. We have described the role of mRNA regarding chemoresistance in section 2 and Table 1.
Comment: Line 141, word of choice “easing”
Revision: Thank you. We have changed the word to “reducing”.
Comment: Line 153, what does the accumulation of ROS do and how is it related to killing tumor cells?
Revision: Thank you for your comment. We have clarified how accumulation of ROS cause DNA damage and kills tumor cells.
Comment: Line 332, tumor-inhibiting capability, additive effect, or synergistic?
Revision: Thank you for your comment. We have clarified the information and it is synergistic.
Comment: Line 333, rephrase, inhibition of EHMT2, a histone lysine methyltransferase, was shown
Revision: Thank you. We have rephrased it as suggested.
Comment: Line 335, “activity” is too broad, be specific eg. kinase activity, binding kinetics.
Revision: Thank you for your comment. We have changed the word “activity” to “enzymatic activity”.
Comment: May move 335 to line 332 as an introduction
Revision: Thank you. We have moved the line as suggested.
Comment: Lin 434, at the end of this section, may discuss the off-target resistance of these inhibitors. Check out Alissa J.Coopers et al. recent review on this topic in nature reviews clinical oncology
Revision: Thank you for your suggestion. We have discussed the off-target resistance and included the reference as per your suggestion.
Comment: Line 399, no need to use KRAS full spelling here as it was first mentioned in line 44
Revision: Thank you. We have fixed it.
Round 2
Reviewer 1 Report
I greatly appreciate the effort made by the authors in critically reviewing their manuscript and responding positively to the comments made earlier by the reviewer.
The manuscript has undergone a general improvement. However, the paragraph on chemotherapy is still heavy and burdened with some grammatical/linguistic errors.
It is suggested that the authors submit the manuscript, particularly the paragraph on chemotherapy, for linguistic revision to improve readability.
The conclusion paragraph is still insufficiently articulated. In light of the multiple therapeutic approaches described by the authors, the conclusions provide an excellent opportunity to discuss what has been described and to enhance the work done.
Author Response
We really appreciate the reviewers’ careful review of our manuscript and useful comments. To address the points raised in the comments, we have now revised the manuscript. These additions and revisions include the following major improvements: (1) Mechanism of chemotherapy resistance (section 2) was revised for linguistic/ grammatical errors, (2) Conclusion was revised to articulate the promising therapeutic approach and future directions.
Below is a description of our responses to each comment (the original comments are quoted verbatim):
Revisions made in response to reviewer #1's comments:
Comment:I greatly appreciate the effort made by the authors in critically reviewing their manuscript and responding positively to the comments made earlier by the reviewer.
The manuscript has undergone a general improvement. However, the paragraph on chemotherapy is still heavy and burdened with some grammatical/linguistic errors.
It is suggested that the authors submit the manuscript, particularly the paragraph on chemotherapy, for linguistic revision to improve readability.
Revision: Thank you for your valuable comments. We have revised the chemoresistance section for linguistic and grammatical errors.
Comment:The conclusion paragraph is still insufficiently articulated. In light of the multiple therapeutic approaches described by the authors, the conclusions provide an excellent opportunity to discuss what has been described and to enhance the work done.
Revision: Thank you for your suggestion. We have rewritten the conclusion and discussed how we can improve the work already done.
Reviewer 2 Report
The reviewer has gone through the highlighted changes. Although the reviewer's previous comments are mostly addressed, some paragraphs (chemo) are still plagued by grammar/wording issues. The reviewer suggests having the co-author who wrote the rest of this paper edit that paragraph for better wording/styling consistency.
Line 42, contribute not contributes
Line 51-53, awkward sentence. Consider change to “Efforts towards classifying lung cancers based on genetic lesions have greatly helped to guide targeted therapies and improved clinical practice.”
Line 63, understanding….is important, not are
Line 65, onset and in response are redundant, consider changing to underlying the “acquisition of resistance to different lines of therapy”.
Line 81, extra space before the dot.
Line 87, add space before “The”
Line 94- 97, Though…. “,”. “followed” changed to “administrated”. Rewrite this part.
Line 101, is “a” controlled process
Line 131, extra space?
Line 153, which “causes”. Of note, only moderate ROS may promote tumorigenesis (the result of fast proliferation), and excessive ROS and DNA damage kills tumor cells. You need to be careful how you describe it.
Line 164, extra space before “WNT”
Line 308, miR-410
Line 697, ( and (
Line 712, conclusions. The review covered more treatment approaches than drugs and should be adequately discussed.
Author Response
We really appreciate the reviewers’ careful review of our manuscript and useful comments. To address the points raised in the comments, we have now revised the manuscript. These additions and revisions include the following major improvements: (1) Mechanism of chemotherapy resistance (section 2) was revised for linguistic/ grammatical errors, (2) Conclusion was revised to articulate the promising therapeutic approach and future directions.
Below is a description of our responses to each comment (the original comments are quoted verbatim):
Revisions made in response to reviewer #2's comments:
Comment:The reviewer has gone through the highlighted changes. Although the reviewer's previous comments are mostly addressed, some paragraphs (chemo) are still plagued by grammar/wording issues. The reviewer suggests having the co-author who wrote the rest of this paper edit that paragraph for better wording/styling consistency.
Revision: Thank you for your suggestion. We have revised the chemotherapy section 2 for linguistic and grammatical errors.
Comment:Line 42, contribute not contributes
Revision: Thank you for your comment. We have corrected it.
Comment: Line 51-53, awkward sentence. Consider change to “Efforts towards classifying lung cancers based on genetic lesions have greatly helped to guide targeted therapies and improved clinical practice.”
Revision: Thank you for your comment. We have changed according to your suggestion.
Comment: Line 63, understanding….is important, not are
Revision: Thank you for your comment. We have corrected it.
Comment: Line 65, onset and in response are redundant, consider changing to underlying the “acquisition of resistance to different lines of therapy”.
Revision: Thank you for your comment. We have changed according to your suggestion.
Comment: Line 81, extra space before the dot.
Revision: Thank you for your comment. We have fixed it.
Comment: Line 87, add space before “The”
Revision: Thank you for your comment. We have resolved it.
Comment: Line 94- 97, Though…. “,”. “followed” changed to “administrated”. Rewrite this part.
Revision: Thank you for your suggestion. We have revised this part.
Comment: Line 101, is “a” controlled process
Revision: Thank you for your comment. We have resolved it.
Comment: Line 131, extra space?
Revision: Thank you for your comment. We have fixed it.
Comment: Line 153, which “causes”. Of note, only moderate ROS may promote tumorigenesis (the result of fast proliferation), and excessive ROS and DNA damage kills tumor cells. You need to be careful how you describe it.
Revision: Thank you for your comment. We have revised the sentence to make it clearer.
Comment: Line 164, extra space before “WNT”
Revision: Thank you. We have fixed it.
Comment: Line 308, miR-410
Revision: Thank you. We have corrected it.
Comment: Line 697, ( and (
Revision: Thank you. We have fixed it.
Comment: Line 712, conclusions. The review covered more treatment approaches than drugs and should be adequately discussed.
Revision: Thank you for your suggestion. We have rewritten the conclusion and discussed how we can improve the work already done.